# Preeclampsia and Obesity—The Preventive Role of Exercise

**DOI:** 10.3390/ijerph20021267

**Published:** 2023-01-10

**Authors:** Elżbieta Poniedziałek-Czajkowska, Radzisław Mierzyński, Bożena Leszczyńska-Gorzelak

**Affiliations:** Chair and Department of Obstetrics and Perinatology, Medical University of Lublin, 20-954 Lublin, Poland

**Keywords:** preeclampsia, gestational hypertension, obesity, exercise, prophylaxis

## Abstract

Obesity is now recognized as a worldwide epidemic. An inadequate diet and reduced physical activity are acknowledged as the leading causes of excess body weight. Despite growing evidence that obesity is a risk factor for unsuccessful pregnancies, almost half of all women who become pregnant today are overweight or obese. Common complications of pregnancy in this group of women are preeclampsia and gestational hypertension. These conditions are also observed more frequently in women with excessive weight gain during pregnancy. Preeclampsia is one of the most serious pregnancy complications with an unpredictable course, which in its most severe forms, threatens the life and health of the mother and her baby. The early identification of the risk factors for preeclampsia development, including obesity, allows for the implementation of prophylaxis and a reduction in maternal and fetal complications risk. Additionally, preeclampsia and obesity are the recognized risk factors for developing cardiovascular disease in later life, so prophylaxis and treating obesity are paramount for their prevention. Thus, a proper diet and physical activity might play an essential role in the prophylaxis of preeclampsia in this group of women. Limiting weight gain during pregnancy and modifying the metabolic risk factors with regular physical exercise creates favorable metabolic conditions for pregnancy development and benefits the elements of the pathogenetic sequence for preeclampsia development. In addition, it is inexpensive, readily available and, in the absence of contraindications to its performance, safe for the mother and fetus. However, for this form of prevention to be effective, it should be applied early in pregnancy and, for overweight and obese women, proposed as an essential part of planning pregnancy. This paper aims to present the mechanisms of the development of hypertension in pregnancy in obese women and the importance of exercise in its prevention.

## 1. Introduction

Hypertension is diagnosed in about 7.7% of women of reproductive age, and its frequency tends to increase. It has been found that in the last 40 years, its average rate rose by 6% per year [1]. Hypertensive diseases affect 8–10% of all pregnancies. Pregnancy-induced hypertension is observed in about 2–8%, while chronic hypertension is observed in 0.9–1.5% of women [2,3].

Preeclampsia (PE) develops in 1% of all pregnant women and occurs in 1.5% of primiparas [4]. According to data from the WHO (World Health Organization), PE is the cause of 76,000 maternal deaths per year, which is about 16% of all maternal deaths, mostly in developing countries. In developed countries, it is recognized as the most common cause of iatrogenic prematurity [5].

One of the identified significant risk factors for the development of hypertension in pregnancy is obesity. Many scientific societies mention obesity as one of the PE risk factors and thus its indication is essential for implementing PE pharmacological prophylaxis with acetylsalicylic acid (AA). Currently, AA is the only approved pharmacological prevention of preeclampsia [3,6,7,8]. However, this prophylaxis has been shown to be effective primarily for early onset preeclampsia [9]. Given the different pathophysiology, course, and prognosis, it is reasonable to differentiate between early onset PE (<34 weeks gestation) and late-onset PE (34 weeks onward). The course and prognosis are significantly worse for the early onset form, but its cases are significantly fewer than late-onset PE [10]. Extensive research in various populations has shown that the frequency of late-onset PE was significantly and linearly associated with an increasing maternal weight, which has not been confirmed for early onset PE [11,12]. The fact that the highly predominant late-onset PE is associated explicitly with pre-pregnancy overweight and obesity leads us to believe that this form of PE is a consequence of metabolic disorders in the mother. Interventions on the gestational weight gain planned since the beginning of pregnancy have the potential to halve the incidence of late-onset PE in overweight/obese women [13].

Obesity and PE have been shown to share many common pathophysiological features. The importance of obesity-specific conditions in the development of PE is supported by the fact that PE occurs more frequently in women with metabolic impairments such as polycystic ovary syndrome or an insulin resistance [3].

Obesity is widely acknowledged as a threat to public health due to the ever-increasing number of people affected worldwide and its health consequences. Since 1980, the number of people who are obese and overweight has doubled and now up to 39% of the world’s population is overweight or obese [14,15]. By 2030, overweight or obesity will affect nearly 58% of the population [16]. This significant increase in the prevalence of obesity results from a change in the diet, decreased physical activity, and environmental, socioeconomic, and genetic factors. Obesity significantly impacts the health of the population due to the increased risk of several diseases such as diabetes, cardiovascular, and skeletal diseases, some types of cancer, or mental illness [17,18,19,20]. All these conditions adversely affect the individual’s health and functioning in society and generate enormous costs for the healthcare system. An increase in the frequency of obesity and overweight is observed and depends on one’s age and gender, geographical location, technical and cultural factors, and the socioeconomic state [14].

The diagnosis of overweight and obesity can be made using the BMI (body mass index), which is calculated based on body weight and height (calculated by dividing the body weight in kilograms by the square of height in meters; kg/m^2^). The classification of overweight and obesity in adults depending on the BMI value is as follows: normal body weight 18.5–24.9 kg/m^2^, overweight 25.0–29.9 kg/m^2^, and obesity ≥ 30 kg/m^2^. A BMI ≥ 40 kg/m^2^ is considered to be severe obesity [21]. Unfortunately, the BMI does not accurately reflect the percentage of body fat for any given BMI value, which can vary significantly (inter-individual variability) between individuals and shows variability based on age, gender, and ethnicity [22,23]. It has been revealed that the body fat content in representatives of the Asian population is higher than in Caucasians with the same BMI [24]. The terms for obesity as metabolically healthy and metabolically unhealthy could be found in the literature. It is believed that among obese people, up to 40% can be healthy. They are characterized by a normal insulin sensitivity, blood pressure, and lipidogram, and a physiological inflammatory response profile. On the other hand, among people with a normal body weight, metabolic disorders characteristic of obesity, such as an insulin resistance, hypertension, and lipid disorders, are also diagnosed [25].

The cardiometabolic risk is also determined by the location of adipose tissue, primarily in the visceral adipose tissue and ectopic depots (such as the muscle and liver) and in cases of an increased fat-to-lean mass ratio (e.g., metabolically obese normal-weight). These observations suggest that the problem of obesity may also apply to people with a normal body weight [26].

Obesity and overweight also affect women of childbearing age and pregnant women increasingly. It is estimated that in the US, 1/2–2/3 of pregnant women are overweight or obese [27,28]. Maternal obesity increases the risk of cardiovascular diseases, metabolic syndrome, diabetes, cancer and psychiatric disorders, and obesity in offspring (Table 1) [29,30]. Many scientific societies acknowledge it as a risk factor for the development of PE [3,6,7].

It is believed that obesity and metabolic disorders passed on to subsequent generations trigger the mechanism of a vicious circle. For this reason, the prevention and treatment of obesity are one of the priority goals of health care [31,32].

Due to these vital complications for overweight and obese women and their children, prevention is essential. Prophylactic management should mainly involve non-pharmacological measures, including physical activity and an adequate diet, implemented in overweight and obese women before the planned pregnancy. Many benefits of exercise in pregnancy are highlighted, including a lower risk of excessive weight gain and the development of hypertension in pregnancy and gestational diabetes, less frequent preterm births, and a higher ratio of vaginal delivery [33,34]. It has also been shown that physically active women are less likely to experience symptoms of depression during and after pregnancy [35].

This article aims to present the mechanisms of the development of hypertension in pregnancy in obese women and the importance of physical activity in its prevention.

## 2. Pathophysiology of Preeclampsia

Preeclampsia is a series of multi-organ clinical symptoms resulting from vascular involvement. It affects the vessels of the trophoblast/placenta, whose incorrect implantation and, consequently, abnormal function are considered to be the starting point for the development of PE. According to the current criteria by the American College of Obstetricians and Gynecologists (ACOG), PE is recognized as new-onset hypertension after the 20th week of gestation with systolic blood pressure ≥ 140 mmHg or diastolic blood pressure ≥ 90 mmHg, measured on two occasions at least four hours apart, and proteinuria of ≥0.3 g per 24 h or ≥1+ proteinuria, detected by a urine dipstick. The lack of proteinuria does not exclude the diagnosis of PE. In its absence, the new onset of any one of the following: thrombocytopenia (platelet count < 100,000/μL), renal insufficiency (serum creatinine concentration > 1.1 mg/dL or a doubling of the serum creatinine concentration in the absence of other renal diseases), an impaired liver function (raised concentrations of liver transaminases to twice normal concentrations), pulmonary oedema, or cerebral or visual problems together with new-onset hypertension mandates the diagnosis of PE. Another form of hypertension in pregnancy is gestational hypertension (GH) or pregnancy-induced hypertension (PIH) with systolic blood pressure ≥ 140 mmHg or diastolic blood pressure ≥ 90 mmHg, which appears after the 20th week in a previously healthy woman, with no signs of proteinuria or other symptoms characteristic of preeclampsia [3]. Gestational hypertension may precede the development of PE, but on its own, it can also adversely affect the outcome of pregnancy. The only causal treatment for PE is delivery.

The criteria for the diagnosis of PE have evolved. Initially, the classical definition of PE included the combined occurrence of hypertension and proteinuria. In later years, attention was also paid to PE-inducing impairment in the functioning of organs such as the liver, kidneys, and central nervous system, hematological abnormalities, and fetal growth restriction (FGR). Consequently, the results of preeclampsia studies may be difficult to compare due to the heterogeneous diagnostic criteria.

Due to the potential risk of severe complications for the mother and fetus and the unpredictable PE course, it is crucial to implement an effective prevention. Currently, the only pharmacological agent recommended by many scientific societies for the prevention of PE is acetylsalicylic acid (AA) [3,6,7,8,36]. Since PE is diagnosed in women without risk factors and in some women who use AA, with a better cognition and understanding of the pathophysiological processes leading to the development of PE, other prevention options are being considered. These include antioxidants (vitamins C and E), calcium supplements, fish oil, nitric oxide supplements, nitric oxide donors, metformin, folic acid, statins, vitamins, weight loss, and physical activity [37].

The causes and factors determining the development of PE have not been fully understood and explained, primarily since PE can occur in two forms, differing in developmental mechanisms and etiological pathways: symptoms of early onset preeclampsia appear <34 weeks; late-onset PE is diagnosed in more advanced pregnancy [38].

An abnormal trophoblast invasion with an incomplete modelling of the spiral arteries leading to placental hypoxia has been widely accepted as the beginning of changes leading to the development of PE. It results in a release of mediators responsible for endothelial damage, oxidative stress, apoptosis, and syncytiotrophoblast microparticles, which can all have a local and generalized effect [39,40,41].

The most typical feature of PE is an impairment of the endothelial function responsible for a generalized vasoconstriction and a reduction in the organ perfusion. Factors such as obesity and diabetes, through adverse effects on the endothelium, intensify the abnormalities observed in PE development [42]. Changes in the endothelium are expected to be more pronounced in PE than in GH, which may explain the more severe course and a worse prognosis for PE [43].

The healthy endothelium produces many vasoactive mediators. The main vasodilation factors synthesized by endothelium include nitric oxide (NO), prostacyclin I2 (PGI2), the endothelium-derived hyperpolarizing factor (EDHF), bradykinin, histamine, and serotonin. Vasoconstrictors include endothelin-1 (ET-1), angiotensin II (ANG-II), thromboxane A2 (TXA2), prostacyclin H2 (PI), and reactive oxygen species (ROS). The proper functioning of the endothelium requires a balance between vasoconstrictors and vasodilators. It is believed that the disruption of the endothelial function is the result of a limited synthesis and the bioavailability of NO, the influence of inflammation and adhesive molecules, excess ROS released during oxidative stress, as well as endothelium-dependent vasodilation disorder, decreased fibrinolysis, and an enhanced endothelial permeability [44].

It has been assumed that the early form of PE results from an impaired placentation and leads to a fetal growth restriction, while the late form is primarily the result of maternal factors. Staff et al. believed that both forms of PE result from an impaired placental function (stage 1), which precedes the appearance of clinical symptoms (stage 2), but the causes and timing of the development of the two forms differ significantly. According to their hypothesis, in the early form of PE, the impaired placental function is ‘extrinsic’ to the placenta with the incomplete remodeling of the spiral arteries, which occurs in early pregnancy and leads to placental hypoperfusion. The cause of the late form is ‘intrinsic’ and consequently is also responsible for restricting intervillous perfusion. As a result, many active substances responsible for the appearance of clinical symptoms are released from the placenta into the maternal circulation. According to the authors, maternal factors acting at multiple stages of PE development contribute to the increased risk of PE in both forms [45].

The causes of an abnormal trophoblast implantation remain unexplained. One of them is supposed to be maternal immune maladaptation and the inadequate induction of tolerance for the allogeneic fetus [46,47]. PE has been shown to be dominated by an inflammatory-type response, whereas in physiological pregnancy, the anti-inflammatory response prevails [48,49].

The levels of the most important pro-inflammatory cytokines, mainly interferon gamma (IFN-γ) and tumor necrosis factor alpha (TNF-α), IL-1, -2, -6, -8, are elevated, while the concentrations IL-10, which is an anti-inflammatory cytokine, is decreased [50]. PE is associated with higher levels of serum heat shock protein 70 (HSP 70), and the degree of the elevation of HSP 70 correlates with the elevation of circulating pro-inflammatory cytokines. It results in a systemic inflammatory response leading to oedema and extravasation, compounding the insults to the placental, renal, and other organ vascular beds [51,52]. Additionally, preeclamptic placentas showed the increased expression of Toll-like receptors 4 (TLR4), responsible for activating the pro-inflammatory response [53]. The study by Xie et al. has revealed that very high TLR4 levels are characteristic of early onset PE and HELLP syndrome [54].

C-reactive protein (CRP) belongs to the acute phase proteins and is mainly produced in the liver and adipocytes, and it is considered to be an exponent of inflammation. The study by Raio et al. has shown an increase in the CRP and IL-6 levels assessed in normal pregnancy, while in PE, it was significantly more pronounced. The CRP concentrations were positively correlated with the levels of an anti-angiogenic factor, suggesting the involvement of CRP in the pathogenesis of PE and indicating the potential for CRP to be used as an early marker of PE [55]. Similar observations have been made by Hamadeh et al. Their systematic review of 34 papers has indicated a positive association between CRP levels and the development of PE. In addition, in their opinion, the assessment of the CRP levels could be a cheap biomarker for the development of PE, and its concentration > 15 mg/L might indicate the prophylactic use of AA [56].

It has been shown that an excessive pro-inflammatory response in PE results in the increased expression of adhesion molecules: ICAM-1 (Intercellular Adhesion Molecule 1) and VCAM-1 (Vascular Cell Adhesion Molecule 1), and the potent vasoconstrictor endothelin 1. All of these are exponents of endothelial damage [57,58].

Complement system abnormalities have been identified as one of the factors responsible for an abnormal trophoblast development. The Lynch et al. studies have shown that elevated levels of the C3 component, which reflects an alternative complement pathway activation, is a significant risk factor for developing PE and GH [59,60]. Obese pregnant women with significantly elevated C3 concentrations are 8–10 times more likely to develop PE than lean women with normal C3 concentrations [61].

The abnormal implantation of the trophoblast has been recognized as a cause of hypoxia and the development of oxidative stress. This process is accountable for the pathophysiological changes accompanying the development of PE and takes place before the appearance of clinical symptoms [62]. These are supposed to be responsible for shifting the balance between angiogenic and anti-angiogenic factors in favor of anti-angiogenic ones, considered to be one of the essential stages in PE development [63].

The angiogenic factors include the following: the vascular endothelial growth factor (VEGF), the placental growth factor (PlGF), and the transforming growth factor-β (TGF-β) family. The VEGF exerts its effects via binding and activating two cell surface receptor tyrosine kinases, VEGFR-1/Flt-1 and VEGFR-2/KDR, that are presented on endothelial cells [64]. The angiogenic effect of the VEGF is to stimulate the proliferation of endothelial cells and maintain their integrity and enhance the synthesis of plasminogen activators [65]. Another role of the VEGF in pregnancy is to intensify the formation of blood vessels in villous and extravillous trophoblasts [66,67]. The link between oxidative stress and the VEGF has been confirmed. In severe PE, changes in the VEGF levels have been shown to increase the activity of the 5’ adenosine monophosphate-activated protein kinase (AMPK), which is expressed in almost every cell type and tissue [68]. AMPK plays a crucial role in processes such as oxygen regulation, cellular energy homeostasis, and the metabolism and is also involved in angiogenesis within the placenta [69].

It has been suggested that an increase in the AMPK activity is a compensatory mechanism for the imbalance between angiogenic and anti-angiogenic factors under reduced placental perfusion [70].

Another angiogenic factor is a member of the vascular endothelial growth factor family proteins, the PlGF, which enhances the angiogenic effect of the VEGF and improves the endothelial cell adhesion and chemotaxis, and is responsible for non-branching angiogenesis [70,71].

Proteins belonging to the transforming growth factor-β (TGF-β) family exert angiogenic properties, and their levels in PE patients are significantly lower than in healthy pregnant women. They control proliferation and differentiation in most cell types, including endothelial cell growth and angiogenesis, and have anti-inflammatory effects. The expression of the VEGF is enhanced by the TGF-β [72,73].

The anti-angiogenic factors of importance in PE pathogenesis include the VEGF receptors (VEGFR1 and VEGFR2) and soluble endoglin (sEng). VEGFR1 is also known as fms-like tyrosine kinase-1 (Flt-1), and the mitochondrial mechanisms determine its secretion. It has been observed that the inhibition of the mitochondrial electron transport chain significantly reduced the secretion of sFlt-1 (soluble fms-like tyrosine kinase-1) by primary villous cytotrophoblasts cells [74]. In PE, the increased activity of the mitochondrial electron transport chain has resulted in an enhanced sFlt-1 release [75]. The release of sFlt-1 in placental explants has been shown to be induced under the influence of HIF1α (hypoxia-inducible factor α), which is a central mediator of the hypoxic response and seems to be the molecular link between placental hypoxia and the downstream mediators of preeclampsia. Its concentration has been shown to increase under ischemia/hypoxia conditions [76]. It has been found that the concentration of sEng, an extracellular domain of the full-length membrane endoglin, was significantly elevated in pregnant women with PE [77]. Anti-angiogenic factors are thought to be responsible for a systemic endothelial dysfunction, which clinically manifests as hypertension and multi-organ disorders [78].

The administration of sFlt-1 led to the development of hypertension, severe proteinuria, and significant histological changes which are typical of PE [79]. It has been revealed that the circulating level of the PlGF is lower in patients who will be preeclamptic before the increase in s-Flt [80].

This observation has been supported by Govender et al., who confirmed that significantly elevated sFlt-1 levels in pregnant women with early onset PE and high sFlt-1 levels in patients with late-onset PE compared to healthy pregnant women [81]. Similarly, the administration of sEng to pregnant animals increased the arterial pressure and the appearance of proteinuria, but its severity was mild to moderate. In contrast, sFlt-1 resulted in the development of severe hypertension and proteinuria, as well as in the appearance of an HELLP syndrome manifestation (HELLP—hemolysis, elevated liver enzymes, low platelets), which is considered as PE with severe symptoms [82]. Both sFlt-1 and sEng contribute to inhibiting the VEGF and TGF-β1 activity [82,83], which decreases the activity of eNOS (endothelial nitric oxide synthase) and consequently reduces the synthesis and release of the potent vasodilator NO. NO is essential for a normal trophoblast development and implantation [84].

The angiogenic factors, the VEGF and PlGF, have been shown to be implicated in the enhancement of the synthesis of NO [85,86]. Thus, the impaired balance of the angiogenic and anti-angiogenic factors observed in PE influences the synthesis of NO. sFlt-1, by reducing the availability of the PlGF and VEGF, causes a reduction in NO synthesis, which is also disturbed by the ROS released during oxidative stress by the hypoxic trophoblast [87]. In addition, it has been shown that the anti-oxidant heme oxygenase-2 (HO-2) expression is downregulated in PE, resulting in increased oxidative stress and ROS concentrations [88]. These observations made it possible to use the assessment of anti- and angiogenic factors in the prediction of PE. An elevated sFlt-1/PlGF ratio in pregnant women with a suspected early onset PE ratio is associated with worse maternal and fetal outcomes in the following two weeks [89]. The multicenter study in high-risk pregnant women in the second and third trimesters has shown that a lowered sFlt-1/PlGF ratio value allows for the reliable exclusion of the development of PE within one week [90].

The results of the above studies suggest that pharmacological and non-pharmacological agents affecting the reduction in placental sFlt-1 and sEng synthesis may have a beneficial effect on the endothelial function and, if they are safe in pregnancy, their use may be considered for PE prevention [75].

Metalloproteinases (MMs), whose levels are elevated in normal pregnancy, are involved in the extracellular matrix degradation and placental implantation. The invasive potential of extravillous trophoblast cells correlates with the MMP-2 and MMP-9 expression, which is reduced in PE [91]. In an animal model, it has been observed that sFlt-1 reduces the activity of MMPs in the placental vascular wall, while the VEGF transposes this process and facilitates a proper trophoblast implantation [92].

One of the mediators affecting the release of sFlt-1 is the PPARγ (Peroxisome Proliferator-Activated Receptor-γ), also known as the glitazone reverse insulin resistance receptor [88]. The PPARγ is a transcription factor from the ligand-activated nuclear hormone receptor family, which, by controlling the adipogenesis, lipid metabolism, and inflammation processes, is responsible for maintaining metabolic homeostasis. The PPARγ has been the best understood of all the PPAPR isoforms in the amnion, decidua, and villous placenta [93,94].

Early in the placental formation, the PPARγ has been shown to inhibit an extravillous cytotrophoblast invasion and promotes a trophoblast differentiation [95]. Crocker et al. believe that a PPARγ-mediated trophoblast differentiation may account for the increased cell resistance to hypoxia-induced apoptosis [96]. It has been found that PPARγ activating factors (presumable ligands) are present throughout pregnancy [97].

It has been observed that in PE, the PPARγ ligand levels were reduced before the onset of the disease symptoms, suggesting a possible role of PPAR signaling in the pathogenesis of preeclampsia [98].

An experimental study by Rahardjo et al. has shown that the plasma from preeclamptic women down-regulated the PPARγ and stimulated a pro-inflammatory response, expressed as an increase in pro-inflammatory cytokines such as IL-1 α, IL-6, and TNF-α. In addition, women with PE have been detected to have an attenuated PPARγ activation, which may reflect increased pro-inflammatory processes [99]. Animal studies have demonstrated that the administration of a PPARγ antagonist was associated with significantly higher sFlt-1, reduced VEGF levels, and preceded the development of PE [100]. The results of Armistead et al. have suggested that an increased PPARγ activity reduces the sFlt-1 expression in the placenta during the first trimester [101].

An impaired lipid metabolism has been found to play an essential role in PE pathogenesis. Increased total serum cholesterol concentrations in the first and second trimester of gestation have been shown to precede PE development [102]. An abnormal lipid profile characterizes pregnant women with PE: increased concentrations of low-density lipoproteins (LDLs), low high-density lipoprotein levels (HDLs), and increased levels of triglycerides (TGs), as well as increased concentrations of free fatty acids (FFAs) [103,104]. Excess LDLs interfere with the trophoblast implantation by reducing the trophoblast migration and increasing trophoblast apoptosis [105], while FFAs activate PPARs [106].

A disturbed endothelium is characterized by an increased peroxidation of the endothelial lipids and reduced anti-oxidant processes. These phenomena lead to the activation of cyclooxygenase (COX), which results in an increased TXA2 synthesis and an imbalance of TXA2/PGI2 in favor of TXA2 [107,108].

It has been thought that endoplasmic reticulum stress (ERS) may play a role in the pathogenesis of PE, which is exacerbated in conditions that are also risk factors for PE development, such as an abnormal glucose metabolism and oxidative stress. Findings suggest that the role of ERS in the pathogenesis of PE may rely on the enhancement of the release of pro-inflammatory cytokines, anti-angiogenic factors, and trophoblastic apoptotic debris that impair the endothelial function. It has also been found to induce apoptosis, which is increased significantly in the severe, early onset form of PE [109,110,111].

The impaired endothelial function may be expressed in the increased vascular permeability confirmed in pregnant women with PE [112]. There is much evidence that it results from an imbalance between the angiogenic and anti-angiogenic factors [80].

The restoration of a normal endothelial function is enabled by endothelial progenitor cells (EPCs), which are endothelial cell precursors. They are important in the angiogenesis improvement and remodeling of vessels [113]. The significantly lower number and abnormal function of EPCs compared to healthy pregnant women have been demonstrated in PE [114,115]. It has been thought that this may result from inflammation and the influence of anti-angiogenic factors [115,116].

The activation of the renin–angiotensin–aldosterone system observed during normal pregnancy leads to increased concentrations of renin, angiotensinogen, and angiotensin II [117]. This system seems to be inhibited in PE as evidenced by lower concentrations of angiotensin I, angiotensin II, aldosterone, and increased levels of antibodies to the angiotensin II type 1 receptor (ATR1-AA) and renin plasma activity. ATR1-AA, through the stimulation of ATR1, is supposed to be responsible for increased blood pressure. The importance of RAAS in the PE pathogenesis has not been unequivocally proven, apart from ATR1-AA [118,119].

## 3. Obesity and Preeclampsia

Obesity adversely affects pregnancy and obstetric outcomes. The risk of PE is two times higher with a maternal BMI of 26 kg/m^2^ and as much as three times higher with a BMI > 30 kg/m^2^ [120]. Furthermore, it has been found that excessive weight gain during pregnancy is associated with a higher risk of developing PE [121]. Therefore, it seems that one of the methods to prevent PE development should be a weight reduction in women of childbearing age, primarily through a proper diet and regular exercise. It has been supported by the results of the study presented by Magdaleno et al., who found that a weight reduction before pregnancy effectively reduces the PE risk [122].

Pre-pregnancy obesity with normoglycemia has been thought to worsen the obstetric outcomes to a greater extent than hyperglycemia in non-obese pregnant women. The risk of PE developing is influenced by the BMI and body fat distribution; women with an “apple silhouette” are at a higher risk [123].

Women who have undergone PE are more likely to develop hypertension, ischaemic heart disease, myocardial infarction, and thromboembolism in later life, which are also characteristic complications of obesity and metabolic syndrome. In addition, it is believed that it is not hypertension alone in pregnancy but the complex metabolic disorders accompanying PE that are most important in the prognosis for future cardiovascular health. There is also an opinion that obesity and the metabolic status before pregnancy, more than pregnancy or PE itself, may be responsible for the development of cardiovascular disease in the future [122,124].

Adipose tissue is an active endocrine and paracrine organ that synthesizes various substances called adipokines. They are involved in several processes, such as the glucose and lipid metabolism, angiogenesis, inflammation, and blood pressure. Their biological functions may represent a link between an excessive body weight, insulin resistance, atherosclerosis, type 2 diabetes, and pro-inflammatory reaction [125]. An insulin resistance causes the onset of atherogenic dyslipidaemia and the endothelial dysfunction, which is further mediated by the adipokines and free fatty acids released by visceral adipose tissue. A weight loss by obese patients leads to the correction of many metabolic abnormalities [126]. Central obesity precedes the development of an insulin resistance. Excess visceral fat is more closely associated with its development than other types of adipose tissue [127].

### 3.1. Adipokines

Although they are synthesized in adipose tissue, in terms of their structure or function, adipokines are not a homogeneous group. To date, some 600 adipokines have been identified [128]. These include enzymes, cytokines, growth factors, or hormones. Furthermore, some adipokines are secreted by adipose tissue and other endothelial or blood cells [129]. Examples of adipokines are leptin, adiponectin, resistin, TNF-α, IL-6, IL-1α, the plasminogen activator inhibitor 1 (PAI-1), the protein involved in the blood pressure regulation—angiotensinogen, the cholesterol ester transfer protein (CETP), monocyte chemoattractant protein-1 (MCP-1), chemerin, retinol binding protein-4 (RBP4), vaspin, visfatin, progranulin, the enzymes involved in the biosynthesis of steroid hormones, and many others [130]. It has been shown that adipokine concentrations correlate with the BMI [131]. A weight reduction has been demonstrated to reduce adipokines and inflammatory marker concentrations and simultaneously correct the insulin resistance [132]. Among the adipokines that are best understood and appear to be important in developing PE are resistin, leptin, adiponectin, TNF-α, and IL-6 [133,134].

#### 3.1.1. Resistin

The mechanisms determining the action of resistin, a pro-inflammatory factor responsible for an insulin resistance (IR) development in humans, are not fully acknowledged. Its expression appears to positively correlate with an IR as a consequence of obesity [135], but according to Panidis et al., elevated resistin levels appear to reflect an IR rather than being its cause [136].

The effect of resistin on the vasoactive endothelial function is of interest—resistin enhances the endothelin-1 expression in endothelial cells, which may be one of the elements responsible for an increased blood pressure in obesity [137]. A study by Luo et al. has shown that resistin exacerbated ER stress and impaired the eNOS activity, consequently reducing the synthesis of NO. In addition, the resistin-mediated release of ROS and pro-inflammatory cytokines such as TNF-α and IL-1β were enhanced [138]. The expression of resistin in the placental tissue has been demonstrated [139,140]. During normal pregnancy, its concentration is significantly higher compared to non-pregnant women and is observed to increase with the advancement of pregnancy, while the expression of resistin in adipose tissue does not change. Hence, the increased synthesis of resistin in the placenta is supposed to be responsible for the progressive insulin resistance observed even in healthy pregnancies [141,142].

Data on the levels and role of resistin in PE are vague [142,143]. However, a possible indirect resistin involvement in the PE pathogenesis by enhancing inflammation has been postulated—it has been shown to intensify the release of IL-6 and TNF-α [144].

#### 3.1.2. Leptin

Leptin is the adipokine responsible for controlling the energy balance and is produced almost exclusively by adipose tissue and the placenta [145]. The role of leptin is to maintain a stable body weight during periods of an excess food supply. Once leptin binds to receptors in the hypothalamus, neurons stop producing neuropeptide Y, an appetite stimulator. Thus, this hormone reduces appetite and stimulates the sympathetic system. Its role in regulating the sensitivity to insulin has been proposed, but in obesity, it can exacerbate an insulin resistance [133]. Leptin exhibits a pressor effect through the activation of the sympathetic nervous system and increased endothelin-1 biosynthesis on the one hand, and a relaxant effect through the increased expression of eNOS on the other [146,147]. It has been revealed that leptin enhanced the platelet aggregation and induced the ROS formation [148,149]. It has been thought to regulate innate and adaptive immune responses and, due to a structure similar to IL-6, to enhance the pro-inflammatory response [150].

In obese individuals, hyperleptinemia, associated with a low-grade inflammatory condition, has been found to explain the development of autoimmune diseases, reproductive disorders, and pregnancy complications [150,151]. The expected anorexic responses have not been observed in obesity with elevated leptin levels, suggesting a leptin resistance [152].

Pregnancy and obesity are leptin-resistance conditions [153]. A growing number of observations have supported the involvement of leptin in PE pathogenesis. Leptin has been found to be committed in the trophoblast implantation by influencing the trophoblastic growth factors and by activating MMPs [154]. In an animal model, it has been shown that hyperleptinemia, through a sympathetic activation, affected the synthesis of NO, and via an aldosterone-dependent mechanism, it was responsible for increased blood pressure and endothelium dysfunction [155,156]. Additionally, leptin may be implicated in PE pathogenesis due to its pro-inflammatory effects [151]. Its elevated levels in PE may be explained by placental stress increasing the nutrient delivery to the fetus or stimulating the leptin expression through the hypoxic placenta [157,158]. The association of leptin with the development of hypertension in pregnancy was confirmed by Poniedziałek et al. They have observed higher leptin levels in pregnant women with GH. This observation indicates a possible link between leptin and GH development and, given the significantly higher BMI in women with GH, points to hyperleptinemia as a factor linking obesity to GH development [159].

#### 3.1.3. Adiponectin

The primary biological function of adiponectin, which is synthesized almost exclusively in adipose tissue, is to increase the insulin sensitivity and exert anti-inflammatory and anti-atherosclerotic effects [160,161]. It also exhibits angiogenic and antioxidant properties [139,162].

The adiponectin levels, unlike other adipokines, are reduced in overweight and obese individuals and correlate inversely with an IR [163,164]. In normal pregnancy, its concentrations decrease as pregnancy progresses and inversely correlates with the amount of adipose tissue [165].

The synthesis of adiponectin is inhibited by pro-inflammatory cytokines, hypoxia, and oxidative stress. Its decreased levels have been observed in obesity, diabetes t.2, and hypertension [166]. Adiponectin, like other adipokines, affects blood vessels. Its actions include the enhancement of the synthesis and release of NO, the inhibition of the TNF-α-induced expression of endothelial adhesion molecules VCAM1, ICAM-1, and E-selectin, the inhibition of monocyte adhesion to endothelial cells, the inhibition of the macrophage transformation into foam cells and reduction in the lipid accumulation and phagocytic activity of mature macrophages, and the inhibition of the TNF-α secretion by macrophages, which explain its hypotensive and anti-atherogenic effects [167,168]. Despite such properties of adiponectin, the results of studies on its importance in the pathogenesis of PE are inconclusive. Some authors have shown its increased concentration in PE, while others have not confirmed this relationship [169,170,171,172].

### 3.2. Inflammation

According to expert opinion, the inflammatory reaction may be the link between obesity, IR, the development of atherosclerosis, and type 2 diabetes. Following the theory that obesity is associated with inflammation, it is assumed that adipocytes initiate the inflammatory process while macrophages exacerbate it. The infiltration of inflammatory cells into adipose tissue induces changes in the lipid metabolism in the adipocytes and biosynthesis of pro-inflammatory cytokines. High concentrations of pro-inflammatory cytokines result from obesity, especially the visceral type. It is believed that the increased biosynthesis of these cytokines by adipocytes results in generalized inflammation, endothelial dysfunction, hemostatic dysfunction, and an IR [173]. There is also evidence that pro-inflammatory adipokines secreted in excess in obesity may be a factor linking inflammation to PE and GH [174,175]. Adipose tissue is an essential source of TNF-α, which levels are directly proportional to the BMI and plasma insulin concentrations. A reduction in body weight is associated with its reduced concentrations [164]. TNF-α has auto- or paracrine effects: it directly induces an IR by inhibiting an insulin signal transduction and indirectly by increasing the free fatty acids levels [133,176]. In addition, it has been shown to increase the resistin levels and release IL-6 from adipose tissue [177]. TNF-α is a major cytokine that induces the endothelial dysfunction by activating the expression of adhesion molecules in endothelial and vascular smooth muscle cells. It inhibits eNOS, which results in a significant reduction in NO synthesis [167]. The adverse effects of TNF-α on endothelial cells are compounded by an existing insulin resistance [178,179]. Furthermore, TNF-a in obesity disrupts endothelial integrity by inducing endotheliocyte apoptosis, which is prevented by insulin [180].

IL-6 is a pro-inflammatory adipokine, and its levels have been demonstrated to correlate with obesity, an impaired glucose tolerance, and an IR. Conversely, a weight reduction is associated with the normalization of its concentrations. IL-6 has been reported to enhance the IR. The following mechanisms have been postulated: the attenuation of insulin signaling in peripheral tissues (the effect on the insulin receptor), adverse effects on the leptin receptor function, and the reduction in the secretion of adiponectin [181,182]. IL-6 also adversely affects the endothelium. Wassmann et al., have revealed that it increased the angiotensin II-induced ROS synthesis, leading to an endothelial dysfunction [183]. A study in animals with induced diabetes has demonstrated that TNF-α and IL-6 exacerbated oxidative stress and inhibited the eNOS activity, thereby contributing to an endothelial dysfunction [184].

The CRP levels have been demonstrated to correlate positively with obesity, blood pressure, the triglyceride levels, fasting blood glucose, and insulin sensitivity. It enhances the expression of adhesion molecules, angiotensin II receptors and the monocyte chemotaxis protein MCP-1, facilitates the LDL uptake by macrophages, and induces the endothelin1 synthesis and vascular smooth muscle cell migration and proliferation [185]. It also inhibits the NO biosynthesis and angiogenesis by affecting eNOS [186].

Given these different biological actions, it is unsurprising that elevated plasma CRP levels represent one of the strongest and independent prognostic factors for the development of cardiovascular disease in obesity [187].

Adipose tissue has been shown to express components for alternative complement pathways, including C3. Elevated C3 levels via TLR4 enhance the synthesis of pro-inflammatory cytokines [188]. It indicates the local activation of this system in obesity [189].

### 3.3. Nitric Oxide

The bioavailability of NO, considered one of the primary exponents of the endothelial function, is reduced in obesity [190]. It is the result of decreased eNOS activity [191]. Conversely, the synthesis and release of NO mediated by the inducible isoform of NO synthase (iNOS) is increased in obesity, which is considered one of the causes of an IR [191,192]. Intriguingly, endothelial dysfunction has been shown to affect only obese individuals diagnosed with an insulin resistance [193]. This relationship may explain so-called healthy obesity.

### 3.4. Dyslipidemia

Lipid metabolism disorders such as an increase in triglyceride (TG), low-density lipoproteins (LDL), residual lipoproteins, apolipoprotein B, free fatty acids (FFAs) levels, and a decrease in the HDL cholesterol levels are present in most obese patients. They are closely associated with an increased risk of atherosclerotic cardiovascular disease [194].

### 3.5. PPARs

The process of adipogenesis is supervised by the peroxisome proliferator-activated receptors (PPARs) present in three major isoforms α, β/δ, and γ. They are the nuclear transcription factors involved in the lipid and glucose metabolism. PPAR-γ is an isotype in adipose tissue, the colon, kidney, and skeletal muscle. The primary function attributed to PPAR-γ is the regulation of adipogenesis and the process of the differentiation of preadipocytes into mature adipose tissue cells. By regulating the FFA levels, PPAR-γ plays a role in sensitizing cells to insulin. PPAR-γ are also involved in the proliferation, differentiation, and survival of cells other than adipocytes; PPAR-γ ligands have been described to inhibit the growth and induce the apoptosis of cells, including cancer cells [195]. It has been reported that the endogenous ligands of PPARs are unsaturated fatty acids, eicosanoids, components of oxidized low-density lipoproteins (LDLs) and very-low-density lipoproteins (VLDLs), and the derivatives of linoleic acid [196,197]. PPARs have anti-inflammatory properties. They repress the target genes of the transcription factors such as nuclear factor-κB (NF-κB), the nuclear factor of activated T cells, activator protein 1, and the signal transducers and activators of transcription in a signal-specific manner [198]. Diet-induced obese mice have shown that the activation of PPAR-γ reduced T-lymphocyte-dependent inflammation within adipose tissue and an IR development [199].

### 3.6. Insulin Resistance

Obesity can cause an insulin resistance and hyperinsulinemia, but a mechanism leading from an insulin resistance to obesity is also considered [133]. One of the elements linking these conditions is oxidative stress, a primary phenomenon in an IR development [200]. The excess of visceral adipose tissue is associated with adipocyte IR and increased lipolysis, which, consequently, due to the increased FFAs supply and accumulation, exacerbates the IR and is responsible for the development of hyperinsulinemia and hyperglycemia [201]. It is believed that the most crucial mechanism causing an IR is the disruption of the cellular insulin signal transduction in target tissues. In adipocytes and skeletal muscle, reduced insulin binding to its receptor has been observed. A reduced glucose transport protein (GLUT4) expression and impaired insulin gene expression under the influence of elevated glucose concentrations and FFAs may also be one of the causes of an IR and impaired glucose transport in adipocytes [202]. The pro-inflammatory kinase-b IkB (IKK-b)/NF-kB pathway reported in obesity interferes with the signal transduction through the insulin receptor [203,204].

### 3.7. D-Chiro Inositol Phosphoglycan

D-chiro inositol phosphoglycans (DCI) are the second messengers of insulin. It has been observed that the DCI concentrations in the placentas and fluids of patients with preeclampsia are significantly higher than in healthy women. According to Scioscia et al., the increased synthesis of DCI in preeclampsia may be a compensatory mechanism for an insulin resistance. This theory is supported by a study showing that an early inositol supplementation during pregnancy may reduce the incidence of gestational diabetes [205]. On the other hand, the observed excess of DCI in preeclampsia may mimic the hyperinsulinemia that accompanies an insulin resistance and which is associated with endothelial damage. Hence, the dynamic control of the vascular function may be altered by excess DCI. Ultimately, it may result in a reduced NO bioavailability, increased peripheral resistance, and hypertension [206]. The lipidic form of DCI in urine appears to be a promising predictor of preeclampsia because it is detected before the onset of its clinical signs [207].

### 3.8. Obesity and Hypertension

Researchers have presented a growing body of evidence linking the prevalence of hypertension to obesity and an insulin resistance. Obesity has been recognized as a risk factor for hypertension [208,209]. The prevalence of hypertension is continuously proportional to the BMI regardless of gender and is 15% at a BMI < 25 kg/m^2^ and as high as 40% in those with a BMI ≥ 30 kg/m^2^ [210].

The increase in the occurrence of hypertension is mainly related to abdominal obesity, as an increased waist circumference is an independent and the most important predictor of its development [211]. The etiological factors of hypertension include hemodynamic disturbances accompanying obesity, an increased peripheral vascular resistance associated with an impaired endothelial cell function, an insulin resistance, and the adipokines’ influence [212].

Under physiological conditions, insulin causes vasodilatation by stimulating the synthesis of NO and thus influencing the capillary recruitment. It “paves the way” to target tissues and organs. In contrast, a damaged endothelium does not respond adequately to the influence of insulin, which is expressed in a reduced NO synthesis and results in an impaired insulin action in target organs. In diabetes, a reduced synthesis and the abnormal response of the vascular wall to NO have been demonstrated. In this context, it is not surprising that drugs that improve the endothelial function simultaneously cause an increase in the sensitivity to insulin. Research on the action of insulin has revealed that this hormone has both metabolic and vascular effects. It is mediated by the enzyme phosphatidylinositol-3 kinase (PI3K), which is responsible for a glucose utilization in peripheral tissues and the synthesis of NO in endothelial cells. The vascular effect of insulin is catalyzed by mitogen-activated protein kinase (MAPK) and is expressed in the increased synthesis of the PAI-1 and ET-1 in endothelial cells. In IR, the pathway mediated by PI3K in skeletal muscle and endothelium is blocked, while the pathway catalyzed by MAPK remains active [213,214].

The consequences of endothelial cell damage in IR are as follows: a reduction in the NO and PGI2 synthesis and release, an increase in the ROS, TXA2, ET-1 synthesis, an increase in the pro-inflammatory cytokines and growth factors synthesis and lipid peroxidation, and an increase in the expression of adhesion molecules and changes in the blood vessel wall structure. The reduced synthesis of NO by endothelial cells results in vasoconstriction, an increased vascular wall smooth muscle cell migration, and accelerated platelet aggregation, and coagulation processes. Thus, endothelial damage is a common link between an IR and the development of hypertension [212,215]. Increased FFAs levels and hyperglycemia, mediated by oxidative stress, also contribute to a decreased NO synthesis and impaired endothelial function [216,217].

Hyperinsulinemia enhances the growth, migration, and proliferation of vascular smooth muscle cells and thus leads to the hypertrophy of the vessel wall, the narrowing of the vessel lumen, and an increased peripheral resistance, which results in a hypertension development [218].

Studies point to adipose tissue as an essential source of angiotensinogen and angiotensin II, angiotensin-converting enzyme (ACE), and angiotensin II receptors AT1 and AT2. The expression of angiotensinogen increases in obesity and correlates with the waist-to-hip ratio (WHR) [219].

Pre-pregnancy obesity is an essential factor in PE development as obese pregnant women have been found to have endothelial dysfunction, an abnormal immune function, and often elevated blood pressure values from early pregnancy [220]. In addition, the importance of obesity-specific metabolic disturbances in the development of PE is also supported by the fact that PE is more frequently observed in women with metabolic impairments such as polycystic ovary syndrome or IR [221]. Even increasing the IR during a normal pregnancy can be considered one of the risk factors for the development of PE [222].

The common elements linking PE and obesity are present in Figure 1.

## 4. Vascular Effect of Exercise

Many studies have indicated the importance of specific molecules produced by the placenta and other organs in placental and fetal development. They are named after the organ they are released by: placenta—placentokines, muscle—myokines, and adipose tissue—adipokines [223,224,225,226,227].

Their synthesis, release, and function may be influenced by external factors, including exercise [228]. All cytokines released under the influence of exercise, irrespective of the site of synthesis, are generally referred to as exerkines (exercise + cytokines) [229].

### 4.1. Exerkines

It has been well-documented that physical activity is one of the recommended therapeutic and preventive measures for obesity and hypertension [128]. Physical activity involves the muscular system, which controls the metabolism through the production and expenditure of energy. Exercise metabolic and vascular effects may partly depend on the action of exerkines [224,230]. Muscles are also active endocrine secretory organs synthesizing myokines. These substances not only act within the muscles in an auto- and paracrine manner but also affect the metabolism of other tissues and organs. Many such compounds have been detected, but only a small number have been well acknowledged. These include apelin, irisin, myonectin, decorin, musclin, IL-15, IL-6, the fibroblast growth factor 21 (FGF-21) and brain-derived neurotrophic factor, and proteins belonging to the natriuretic peptide family. Their concentrations increase significantly in the response to exercise and their role is not only limited to the control of the muscle metabolism, but they also modulate the systemic metabolism and affect the vessels and circulatory system [224,229,231].

#### 4.1.1. Apelin

Apelin is an adipokine present in several isoforms differing in the site of occurrence and biological potency. It is synthesized mainly in adipose tissue and the endothelium, brain, heart, kidney, liver, and gastrointestinal tract. Apelin exerts its action after binding to the angiotensin II J receptor (APJ), and its synthesis and tissue expression correlate closely with obesity and an IR [232].

Insulin is the most important regulator of the expression of apelin. It is also influenced by hypoxia and adiposity. In animal studies, apelin has been shown to increase the uptake of peripheral glucose and improve the glucose metabolism, which suggests its significance as an insulin-sensitizing adipokine [233,234]. Studies have shown that apelin may have vasodilatory or vasoconstrictive effects depending on the localization of the APJ receptor. After binding to the APJ in endothelial cells, it causes the release of endothelium-derived relaxing factors, including PGI2 and NO [235]. Apelin binding to the APJ in a muscle has vasoconstrictive effects [236]. Another investigation has suggested an additional role for apelin in protecting endothelial cells from diabetes-induced damage and in the process of angiogenesis. It has been thought to reduce the apoptosis and molecule expression via APJ-activated NF-kB pathways [237].

Research on the importance of apelin in pregnancy is scarce. The release of apelin, which significantly affects the fetal and placental development by increasing the oxidative metabolism, has been shown to increase in response to exercise [226,238]. Apelin has not been reported to be related to the IR observed in the third trimester, and no correlation between the apelin and insulin levels in pregnant women has been found [239]. The possible role of apelin in PE pathogenesis has been highlighted, although the data are incomplete and inconsistent. The expression of apelin and blood concentrations increase in preeclamptic placentas [240]. However, other studies have pointed to a down-regulation of the apelin/APJ system in hypertensive disorders of pregnancy [241,242].

In an animal model of induced PE, the administration of apelin has been shown to reverse the increase in arterial pressure and favorably affect fetal growth and survival. The authors believe that it could result from the beneficial effects of apelin on the eNOS/NO signaling pathway and the prevention of oxidative stress development [243].

#### 4.1.2. Irisin

Irisin is one of the best-studied myokines. Outsides the muscles, which are responsible for approx. 70% of irisin synthesis, it is also produced in adipose tissue, the nervous system, as well as in the ovary and placenta [244,245,246].

During exercise, it is synthetized and released through the stimulation of the peroxisome proliferator-activated receptor gamma coactivator 1-alpha (PGC-1α) and proteolytic cleavage of fibronectin type III domain-containing protein 5 [247]. In the opinion of many authors, irisin seems to be the central mediator of the beneficial effects of exercise [248,249,250].

Irisin is one of the essential factors regulating energy processes, as it is responsible for an exercise-dependent energy expenditure. As a result of an increase in the total energy expenditure controlled by irisin, the body weight is reduced, and the insulin sensitivity and glucose metabolism are improved. Therefore, irisin is postulated to have a protective effect against the development of obesity-related diseases such as diabetes, an abnormal lipid metabolism, and cardiovascular disease [251,252,253].

Irisin has been observed to benefit the vascular endothelium and cardiovascular system by reducing oxidative stress and relaxing vessels [248,254]. It has been found that irisin is responsible for vasorelaxation in a dose-dependent manner via endothelium-dependent and independent mechanisms [255,256]. A study conducted by Han et al. in an animal model has shown an improved endothelial function under the influence of irisin, which is the effect of activating the AMPK-eNOS signaling pathway [254].

Some drugs increase irisin concentration and thus have a beneficial effect on the endothelial function. Among them are polyunsaturated fatty acids, insulin, metformin, fenofibrate, and melatonin [257].

These agents are also being studied for their potential use in PE prevention [37].

#### 4.1.3. Interleukin 6

Many studies have shown that the exercise-dependent IL-6 release is influenced by multiple mechanisms, including the enhancement of the mobilization of intramuscular triacylglycerol, fatty acid oxidation, and the translocation of GLUT4 from the cytosol to the membrane. It has been shown that exercise is followed by the increased synthesis of the classic pro-inflammatory cytokine IL-6 [258]. IL-6 has also increased the insulin sensitivity in the muscle via AMP-activated protein kinase [259,260,261]. Thus, physical activity has been found to improve the vascular effects of insulin and, consequently, lead to a decrease in the RR and heart rate [262].

### 4.2. Exercise Impact on Preeclampsia Pathophysiology

It has been reported that exercise before pregnancy and continued or started during early pregnancy significantly reduces the incidence of its complications, including hypertensive disorders [263]. By affecting several mechanisms involved in or specific to the PE pathogenesis, exercise may be an important element in preventing the development of hypertensive conditions in pregnancy [264].

However, there is little information on how exercise could reduce the risk of developing hypertensive disorders during pregnancy. Studies have been conducted on human and animal models, and their results are inconclusive [264,265,266].

The increased risk of developing PE in pregnant women with obesity and an IR is primarily due to an impaired endothelial function and intravascular inflammation [267,268]. It is deemed that physical activity, both before and during pregnancy, can modify various elements that finally lead to the development of PE. Its effects on the placental development, reduction in oxidative stress, and the pro-survival response, which is thought to be responsible for maintaining or improving the endothelial function, might be considered [266]. Physical exercise has a beneficial effect on the above changes and, through its effect on the control of weight and reduction in hormonally active adipose tissue, it could reverse the metabolic abnormalities characteristic of obesity. Additionally, normalizing placental angiotensin II type 1 modulates the RAA system’s function, which consequently might inhibit PE development [269].

#### 4.2.1. Placenta Development

Physical activity has been shown to benefit placental development, with favorable effects on the trophoblastic, endothelial, and stromal cell proliferation [270]. According to Clapp et al., exercise increases the blood flow through the muscle and skin and induces short-term hypoxia within the placenta. As a result, the synthesis of HIF1 is stimulated, and the release of the VEGF and the process of angiogenesis are enhanced [271].

However, the results of studies on the effect of exercise on the placental size are inconclusive. Some authors believe that the placenta of women who exercise during pregnancy have a greater mass and is characterized by an increased functional volume, which translates into a more extensive exchange area [272]. Conversely, many studies have indicated that the placentas of exercising women are smaller than non-exercising women, which is influenced by the intensity of exercise. According to the authors of these studies, however, this does not entail clinically relevant effects [273,274].

However, it should be emphasized that numerous studies suggest that the placental weight-to-birth weight ratio, an important exponent of placental efficiency, does not change under physical activity [273,275]. The inconclusive results of the research may be due to the very different and hardly comparable conditions of the studies in terms of the time, the intensity and type of exercise, and the studied groups [265,266,271,272,276,277].

The proper development of the placenta depends on the predominance of angiogenic factors over anti-angiogenic factors. Studies in animal models and humans have shown the beneficial effect of exercise on the balance between angiogenic and anti-angiogenic factors. The disruption of this balance has been a recognized element in the PE development. Physical activity has been reported to be one of the determinants responsible for the increase in the PlGF levels and the significant reduction in sFLT1 concentrations [272,278]. The research by Genest et al. has confirmed the beneficial effects of exercise on shifting the balance towards angiogenic factors by reducing the expression of sFlt1 and, consequently, on a normal placental development [279].

The study of Gilbert and al. in an animal model has revealed that by reducing sFLT-1, increasing the VEGF levels, and decreasing oxidative stress, exercise abolished the hypertension provoked by a reduced uteroplacental perfusion [265]. Exercise also positively affects the concentrations of heat shock proteins. HSPs have cytoprotective effects and are called “molecular chaperones” as they protect other proteins from the effects of adverse conditions. Gilbert et al. assessed the effects of exercise on the expression of several cytoprotective and pro-angiogenic molecules, such as HSPs and VEGF, in pregnant and non-pregnant rats. Both the HSPs and VEGF expression in the placenta of exercising pregnant rats was significantly higher compared to inactive animals. In addition, the animals showed a more enhanced endothelium-dependent vascular relaxation. In the authors’ opinion, as a result, exercise before and during early pregnancy, which enhances the expression of cytoprotective and pro-angiogenic molecules, may reduce the placental and endothelial dysfunction and thus prevent some pregnancy complications, including PE and GH [265].

The activation of AMPK is responsible for an increased VEGF synthesis [280,281]. Exercise has been shown to increase the AMPK activity in the skeletal muscle and metabolic pathways controlled by this enzyme in a load-dependent manner [282]. Thus, the stimulation of AMPK by pharmacological (e.g., metformin) and non-pharmacological agents (physical activity), which induces a shift in the balance in favor of the angiogenic factors, may justify their use in the prevention of PE [283]. The study by Hardy et al. has yielded equivocal results for different anti-angiogenic agents. On the one hand, the authors have observed increased mRNA activity and concentrations of the VEGF and its receptor in the placenta of exercising women, but on the other, no such relationship has been found for the PlGF concentrations and its mRNA expression [284]. A study by Bhattacharjee et al. in a small group of pregnant women who started the light-to-moderate intensity exercise from the 16 to 20th week has revealed no effect on the HIF1α expression, the VEGF levels, and ER stress or oxidative stress. However, the authors have demonstrated a significant increase in the mRNA expression and angiogenin levels, stimulating the formation of new blood vessels [285].

#### 4.2.2. Oxidative Stress

Oxidative stress has been found as a significant element in PE development. Increased concentrations of ROS and the decreased activity of antioxidant defense enzymes such as catalase, glutathione peroxidase, and superoxide dismutase have been reported [286,287].

In response to an incredibly intense workout, there is a transient increase in oxidative stress, which activates the antioxidant defense mechanisms. It has been observed that exercise was associated with an increase in the superoxide dismutase and glutathione peroxidase activity in the muscle, liver, and plasma [288]. Another defense against oxidative stress is the exercise-induced reduction in the lipid peroxides levels [289]. Physical activity has been reported to increase the number of mitochondria in the muscle, which is responsible for enhancing the resistance to oxidative stress [290].

#### 4.2.3. Inflammation

Abnormalities in the immune response and the predominance of pro-inflammatory processes are thought to be responsible for an impaired trophoblast implantation and subsequent PE development. It has been shown that although very high-intensity physical activity results in inflammation, regular exercise induces an anti-inflammatory response [291]. A study by van Poppel et al. in overweight and obese pregnant women has demonstrated that moderate-to-vigorous exercises were associated with a significant increase in pro-inflammatory cytokines such as IL-6, TNF-α, and IL-1β. In contrast, light exercise was associated with an increase in IL-10, an anti-inflammatory cytokine. The authors of this study speculate that IL-6 via an induced lower first-phase insulin response is thought to provide a euglycemic state in exercising women. Elevated concentrations of other cytokines appear to be associated with obesity [292]. The research by Bjørnstad et al. has revealed that training was accompanied by a decrease in the soluble CD40 ligand and P-selectin concentrations, which probably reflects an attenuated platelet-mediated inflammation. These studies have found no changes in the concentration of other inflammatory molecules such as the TNF-α, MCP-1, and VCAM-1 during exercise [293]. In contrast, Adamopoulos et al. have observed, as a result of exercise, a decrease in the pro-inflammatory markers of an endothelial injury, such as the granulocyte-macrophage colony-stimulating factor (GM-CSF), MCP-1, ICAM-1, and VCAM-1. It may explain the mechanism for the beneficial effects of physical activity on the endothelium [294]. Another possible mechanism by which physical activity may modulate the immune response is the inhibition of TLR 4 and the control of the immune response [295].

#### 4.2.4. Endothelial Function

The main hallmark of PE is endothelial dysfunction. Physical activity is believed to prevent or partially reverse it by activating anti-inflammatory mechanisms and reducing or abolishing oxidative stress. One of the critical features of a healthy endothelium is the adequate synthesis of NO, for which enhanced shear stress is supposed to be responsible. Shear stress is a response to sustained exercise and results from the tangential force exerted by the blood flow on the endothelial surface [296].

Under its influence, the increase in the endothelial cell proliferation and eNOS activity and the reduction in the oxidative stress and inflammation have been observed [297,298,299]. Shear stress increases the synthesis and release of NO, leading to vasodilation and an endothelial function improvement [300,301]. This mechanism explains the beneficial effect of exercise on lowering blood pressure and justifies the potential benefits of physical activity in preventing PE. Under the exercise, vasodilation and angiogenesis have been observed, significantly increasing the endothelial surface area and enhancing endothelial NO and PGI2 synthesis [302].

However, the effect of shear stress on placental circulation is not fully understood. Ramirez-Velez et al. have shown an increase in the eNOS activity and NO release within the placenta in exercising women [303].

The effects of exercise on the pathophysiological elements of PE development are shown in Figure 2.

## 5. Exercise in Pregnancy

Physical activity is defined as any movement that is associated with an increase in physical expenditure. Sport is any planned, structured, repetitive activity undertaken to improve or maintain fitness [304]. The goal of promoting physical activity among women in a healthy pregnancy is to prevent the development of its complications, which include hypertension and gestational diabetes, and the beneficial effect on the subsequent health of the offspring. It also aims to prevent cardiovascular diseases and metabolic disorders in the future. Physical activity during pregnancy is also intended to reduce excessive pre-pregnancy weight in overweight and obese women and limit excessive weight gain during pregnancy [305].

### 5.1. Exercise and Weight Gain during Pregnancy

Excessive weight gain during pregnancy, foremost in overweight and obese women, is associated with a high risk of complications [306]. A study of 265,270 births has revealed that pregnancy in an obese woman with excessive gestational weight gain is associated with the highest risk of developing complications such as PE, GH, GDM, and LGA (large for gestational age) (RR 2.51, 95% CI 2.31–2.74). It was estimated that 23.9% of all pregnancy problems are attributable to maternal obesity/overweight. The authors of this report believe that one of the elements of preventing these pregnancy complications should be the reduction in the pre-pregnancy body weight and the restriction of weight gain during pregnancy, which can be achieved through adequate physical activity and diet [307].

Numerous studies have shown a negative correlation between reported physical activity and excessive weight gain during pregnancy, as defined by the IMG (Institute of Medicine Guidelines) [308,309,310]. Systematic reviews have indicated that weight gain during pregnancy in women in the physically active group was approximately 1 kg less than in non-exercising women [309,310,311,312,313,314,315,316].

Exercise seems to be more effective in reducing weight gain in obese than overweight women [314], although the meta-analysis of 15 RCTs (*n* = 2915) has shown that age, race, or the BMI before pregnancy did not significantly affect the association of weight gain with physical activity. The effect of exercise on weight gain was most pronounced among normal-weight women compared to overweight and obese women [317].

The authors of the i-WIP Collaborative Group report, which included data from 33 studies and 9320 women, have found that exercise and diet reduced weight gain during pregnancy by an average of −0.70 kg regardless of the BMI, age, childbearing, race, or other chronic medical conditions [318].

A meta-analysis ultimately involving 49 RCTs (*n* = 11,444) on the effects of diet alone, exercise alone, and combinations of diet and exercise on the risk of excessive weight gain showed that these interventions reduce it by an average of 20% compared to standard care (RR 0.80, 95% confidence interval (CI) 0.73 to 0.87; *n* = 7096). No intervention was superior to the others—either exercise in any form, supervised or unsupervised, or combined with diet or diet alone—as all of them were effective [310]. However, according to Craemer et al., an appropriate diet seems more effective in reducing excessive weight gain as the sole intervention, or in combination with exercise, than exercise alone [319].

Most publications on weight gain refer to the IOMs recommendations, summarized in Table 2 [308].

In addition to controlled weight gain during pregnancy, a weight reduction between pregnancies is equally important, especially for overweight and obese women. It seems, however, that classes of obesity should be considered [11,12].

A weight reduction has been shown to significantly reduce the risk of complications in subsequent pregnancies, such as hypertensive conditions, GDM or LGA [320]. Indeed, the lowest risk of various complications of pregnancy has been found for women who are of obesity class I (BMI 30–34.9 kg/m^2^) at a gestational weight gain of 5–9 kg, class II (BMI 35.0–39.9 kg/m^2^) at 1–5 kg, and for class III (BMI ≥ 40 kg/m^2^) who experienced no weight gain [321,322].

### 5.2. Exercise and Preeclampsia Risk

Consistent with experimental studies suggesting the beneficial effects of exercise on angiogenesis, vascular endothelium, and metabolic changes, physical activity appears to be an attractive option for PE prevention. Several observational and clinical studies, randomized controlled trials, and meta-analyses have been published on this topic, but their results are ambiguous.

The electronic database PubMed has been searched using keywords such as “exercise” and “preeclampsia”, which resulted in finding 237 articles published within the last ten years. Unfortunately, only a tiny proportion of them attempted to clarify whether exercise could indeed prevent the development of hypertension in pregnancy. Extending the search of the Pubmed database with the keyword “meta-analysis”, 28 papers were found, of which only 14 provided information on the effect of exercise on the PE/GH risk. Only articles available in English were considered.

The results of published meta-analyses on the association of physical activity with the incidence of hypertensive conditions in pregnancy are inconclusive and are summarized in Table 3.

The study of Muktabhant et al. has found no beneficial effect of diet, exercise, or both interventions in reducing the PE risk (RR 0.95, 95% CI 0.77 to 1.16; *n* = 5330), but hypertension in pregnant women in the intervention group was diagnosed less frequently than in the control group (mean RR 0.70, 95% CI 0.51 to 0.96; *n* = 5162) [310]. A meta-analysis published by Aune et al. has demonstrated a favorable effect of pre-pregnancy (RR = 0.65; 95% CI: 0.47 to 0.89; 5 studies) and early pregnancy (RR = 0.79; 95% CI: 0.70 to 0.91; 11 studies) physical activity on reducing the PE risk [323]. The results of the study by Di Mascio et al. have indicated that moderate-intensity leisure activities (aerobic dance, cycling, hydrotherapy, and resistance exercises) practiced during pregnancy are associated with a significantly lower incidence of PE compared to non-exercising women (RR = 0.21; 95% CI: 0.09–0.45) [324].

Magro-Malosso et al. have published a meta-analysis of 17 randomized trials involving 5075 women enrolled before the 23rd week of gestation. The authors have observed that physical activity initiated in early pregnancy (aerobic exercise for about 30–60 min. 2–7 times per week) was associated with an overall lower prevalence of hypertensive disease in pregnancy (5.9% vs. 8.5%; RR = 0.70, 95% CI 0.53–0.83; 7 studies, *n* = 2517). These women had a significantly lower risk of developing GH (2.5% vs. 4.6%; RR = 0.54, 95% CI 0.40–0.74; 16 studies, *n* = 4641) compared to women in the control group. However, these authors have not demonstrated the significance of exercise in PE prevention. The PE incidence was similar in both aerobic exercise (without dietary counselling) and non-exercise women (2.3% vs. 2.8%; RR = 0.79, 95% CI 0.45–1.38; 6 studies, *n* = 2230) [325].

A study by Teede et al. of 5332 patients has demonstrated that exercise significantly reduced the risk of PE and GH developing (RR 0.66; 95% CI 0.48–0.90). The authors focused mainly on studies evaluating the effectiveness of structured physical activity, which included a specific training plan conducted under controlled conditions [326].

There are few studies on the effect of the exercise intensity and duration on the risk of developing PE. A meta-analysis published by Aune et al. aimed to assess the dose–response relationship between physical activity and the risk of PE. The authors have found a 28% reduction in the PE risk for every 1 h of exercise per day (RR = 0.72; 95% CI: 0.53–0.99; 3 studies) and 22% for every 20 metabolic equivalents of task (MET)-hours/week increment (RR = 0.78: 95% CI: 0.63–0.96; 2 studies). The maximum risk reduction has been proven for activity lasting 5–6 h per week, and no further reduction was observed with an increasing training time. Furthermore, the authors of this report have observed that exercise performed in early pregnancy linearly reduced the PE risk by 17% for each 1 h/day increment in physical activity (RR = 0.83; 95% CI: 0.72–0.95; 7 studies) and by 15% for every 20 MET-hours/week increment (RR = 0.85; 95% CI: 0.68–1.07; 3 studies) [323].

Thangaratinam et al., evaluated the effect of weight management interventions that included diet, physical activity, or a combination of both. Women were considered physically active if they spent at least 150 min per week on moderate-to-vigorous exercise. The authors have demonstrated that these interventions were effective in reducing the incidence of PE (RR 0.74, 95% CI 0.59–0.92) and most effective in the group of overweight and obese women (RR 0.65, 95% CI 0.44–0.97). Their diet was most potent in significantly reducing the incidence of PE (RR 0.67, 95% CI 0.53–0.85), but exercise and the combination of diet and exercise did not substantially reduce the incidence of PE (RR 0.65, 95% CI 0.44–0.97). Only diet was effective in preventing GH (RR 0.30, 95% CI 0.10–0.88), while overall weight management approaches were not efficient in reducing the GH risk (RR 0.77, 95% CI 0.54–1.1), which was also prevalent among women with obesity (RR 0.70, 95% 0.30–1.16). An essential finding of this study is that there is no adverse effect of diet, exercise, and their combination on the course and outcome of pregnancy [315]. A systemic review of 11 studies evaluating leisure-time physical activity and the PE risk by Wolf et al. has indicated that intensive physical activity before and during pregnancy or at least 4 h per week reduces the risk of PE [327].

On the other hand, a systematic review by Fazzi et al. on the association of a sedentary lifestyle with the course of pregnancy has yielded inconclusive results on GH and PE development [328].

Studies with different results are also available. The i-WIP Collaborative Group has found no difference in the PE rates in women exercising and following a diet versus non-exercising 0.74; 95% CI: 0.42 to 1.33) regardless of their age, BMI, and race/ethnicity [318].

Kasawara et al. published a systematic review of 17 papers on the importance of exercise in preventing PE. According to them, the results of six case–control studies have demonstrated the effectiveness of exercise in the prevention of PE (RR = 0.77, 95% CI 0.64–0.91), while the results of ten prospective cohort studies have not supported this conclusion (RR 0.99, 95% CI 0.93–1.05) [329]. Other authors have not confirmed the efficacy of exercise in PE prophylaxis [309].

The results of the above meta-analyses do not allow firm conclusions to be drawn regarding the effect of exercise on the risk of PE development. It could be explained by the fact that: 1. the investigations were carried out in heterogeneous groups regarding the BMI, risk of developing PE, and GDM diagnosis; 2. the PE diagnosis was most often based on hypertension and proteinuria, but the recognition criteria were not uniform; 3. the effects of exercise performed during different periods of pregnancy, even up to delivery, were analyzed; 4. some studies assessed physical activity based on questionnaires completed by patients; 5. some studies included any physical activity, while others only supervised exercise. Drawing firm conclusions from the numerous studies that analyzed the association of physical activity with the risk of PE and other hypertensive diseases in pregnancy is hampered by the failure of many authors to consider the time spent exercising per week. An additional difficulty is the intensity of exercise. The great majority of the studies published to date have looked at light to moderate-intensity exercise. There is currently no well-documented safety of intensive exercise for maternal health, hence many doctors do not recommend it. In addition, other forms of prophylaxis, primarily acetylsalicylic acid, were not considered in patients at a high risk of PE included in the studies. The question of the efficacy of exercise as prophylaxis for PE development seems to be resolved. The vascular and metabolic effects of exercise, juxtaposing with elements of the pathogenetic chain of PE development, with a particular focus on obesity, make it appear attractive as non-pharmacological and low-cost prophylaxis if included before the completion of a trophoblast implantation. It should be emphasized that the meta-analyses cited did not show any adverse effects of exercise on pregnancy or fetal development.

## 6. Exercise—Recommendations for Pregnant Women

Numerous scientific societies recommend physical activity in a healthy pregnancy, and many pregnant women want to continue or start exercising for the sake of their health and the health of their offspring. Current recommendations do not differ for particular groups of women regarding their BMI, and the type of physical activity recommended to prevent particular complications. Given the beneficial effects of exercise on the health of pregnant women and the favorable impact on the obstetric outcomes, an exercise program of approximately 150 min per week of low-to-moderate intensity is now recommended. According to the ACOG statement, women in healthy pregnancies should exercise 20–30 min daily most days of the week [33]. The Physical Activity Guidelines for Americans recommends 150 to 300 min of moderate-intensity exercise, especially aerobic exercise, most days of the week during pregnancy and in the postnatal period [339]. Other guidelines suggest moderate-intensity training sessions of at least 25 min at least three times a week. Intense workouts lasting more than 40 min are not advocated [340]. Statistics on the percentage of physically active pregnant women are incomplete. According to the data, only 23–29% of pregnant women in the USA and 20.3% in Spain report activity following the above recommendations [341,342]. Even women who actively exercised before conception, after the onset of pregnancy, reduce the frequency and intensity of exercise similarly to post-partum women [343]. Many women forgo physical activity for various reasons, including the fear of adverse effects on pregnancy, and only about 9.6% of pregnant women increase their physical activity [344]. Supervising the intensity of exercise, which should be adapted for pregnant women, seems to be one of the most critical issues. The most straightforward test is the so-called “talk test”. If the pregnant woman can talk freely during exercise, it should be assumed that the intensity is appropriate [33]. Other recommendations suggest that pregnant women achieve an appropriate heart rate ceiling during exercise and these recommendations are inconclusive. The Canadian and American Society recommends exercise at a heart rate of 60–80% of the maximum aerobic capacity [345,346], while the Royal College of Obstetricians and Gynecologists (RCOG) suggests 60–90% of the maximum heart rate (maximum heart rate = 220—age) for pre-pregnancy exercisers and 60–70% for women inactive before pregnancy [347]. Other recommendations suggest achieving a 30% heart rate reserve (HRR; mild) or a 70% HRR (moderate) intensity. The heart rate reserve is the difference between a person’s resting heart rate and maximum heart rate [340]. Ferrari et al., suggested starting an exercise program during pregnancy with a 25 min training 3 times a week with a heart rate of approximately 30% of the maximum heart rate, then gradually increasing the exercise time by two minutes to a maximum of 40 min. per session to reach 60% of the maximum heart rate [348]. Nevertheless, another way of assessing the exercise intensity is the MET. One MET unit indicates the consumption of one kilocalorie of energy by one kilogram of body weight during one hour of rest (quiet sitting) (kcal/kg b.w./h). The MET number indicates by how many times more energy is expended during exercise compared with the energy expended at rest. Walking at a slow pace has a MET value of 2, while jogging and cycling have MET values of 7–8. For pregnant women, exercise corresponding to around 3–4 METs is recommended [349]. The suggested management for implementing exercise in pregnancy is presented in Figure 3 [347,350].

Not all physical activities are advisable for pregnant women. Examples of safe and unsafe exercise during pregnancy are shown in Table 4 [33,348].

Before starting to work out, it is necessary to consult an obstetrician and assess contraindications to exercise during pregnancy. These can be divided into absolute and relative and are provided in Table 5 [33,348].

There is much concern about the possibility of preterm birth due to exercise. Studies have not shown a higher rate of preterm birth among women who exercise during pregnancy than women with a sedentary lifestyle [316,324].

So far, a sedentary lifestyle or bed rest has not been documented to improve the obstetric outcomes in women at risk of a preterm birth but without its symptoms. Hence, limiting physical activity should not be recommended for preventing a preterm birth [33] and as the primary prevention of PE [3].

## 7. Conclusions

Pre-pregnancy obesity is a recognized factor in the development of hypertension in pregnancy. Experimental studies have shown that obesity and PE share many pathophysiological elements, including endothelial dysfunction, a pro-inflammatory type of immune response or oxidative stress. Physical exercise has a beneficial effect on all these conditions, which makes it appear to be, on the one hand, a cheap and universal way to reduce the incidence of obesity among women of childbearing age and, on the other, an attractive element in the prevention of hypertension in pregnancy, also in women of a normal weight. Physical activity is not only about reducing the body weight and decreasing the secretory active adipose tissue mass, it is also about creating the right metabolic environment for the fetus’ development. Unfortunately, clinical studies do not indicate unambiguously regarding whether exercise significantly reduces the incidence of developing preeclampsia and gestational hypertension. It seems to be a result of the fact that the studies were conducted in heterogeneous groups in terms of the time of inclusion in the study (preconceptional period, early pregnancy, and various stages of pregnancy up to delivery) and the type, duration, and intensity of physical exercise. In addition, some of the studies used survey data for the analysis, making the impartiality of the results much more difficult. Furthermore, not all investigations considered the influence of other factors, such as the pre-pregnancy weight, gestational weight gain, age, race, or socio-economic status.

Despite the extensive literature confirming the effectiveness of exercise in preventing preeclampsia, the issue requires well-designed studies that consider the pathophysiology of PE. Presently, the available data indicate that physical activity in pregnancy is safe for the mother and the fetus and hence can be recommended for most pregnant women in healthy pregnancy.

## Figures and Tables

**Figure 1 ijerph-20-01267-f001:**
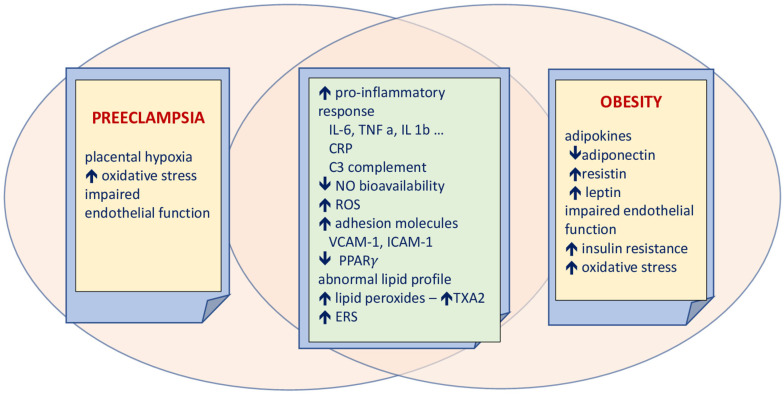
Main features of preeclampsia and obesity. IL-6—interleukin 6, TNF*α*—tumor necrosis factor α, IL-1*β*—interleukin 1 *β*, CRP—C-reactive protein, NO—nitric oxide, ROS—reactive oxygen species, VCAM-1—Vascular Cell Adhesion Molecule 1, ICAM-1—Intercellular Adhesion Molecule 1, PPAR*γ*—Peroxisome Proliferator-Activated Receptor-*γ*, TXA2—thromboxane A2, ERS—endoplasmic reticulum stress.

**Figure 2 ijerph-20-01267-f002:**
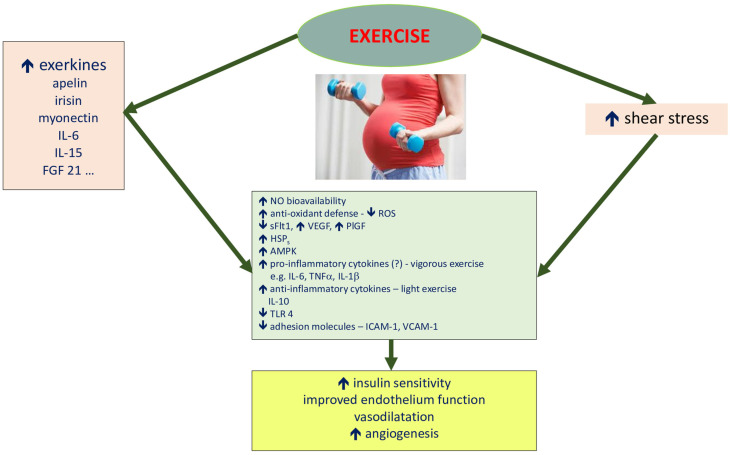
The effects of exercise on pathophysiological elements of PE development. IL-6—interleukin 6, IL-10—interleukin 10, IL-15—interleukin 15, FGF 21—fibroblast growth factor 21, TNF*α*—tumor necrosis factor *α*, IL-1*β*—interleukin 1 *β*; NO—nitric oxide, ROS—reactive oxygen species, sFlt1—soluble fms-like tyrosine kinase-1, VEGF—vascular endothelial growth factor, PlGF—placental growth factor, HSP_s_—heat shock proteins, AMPK—5′ adenosine monophosphate-activated protein kinase, TLR 4—Toll-like receptors 4, VCAM-1—Vascular Cell Adhesion Molecule 1, ICAM-1—Intercellular Adhesion Molecule 1.

**Figure 3 ijerph-20-01267-f003:**
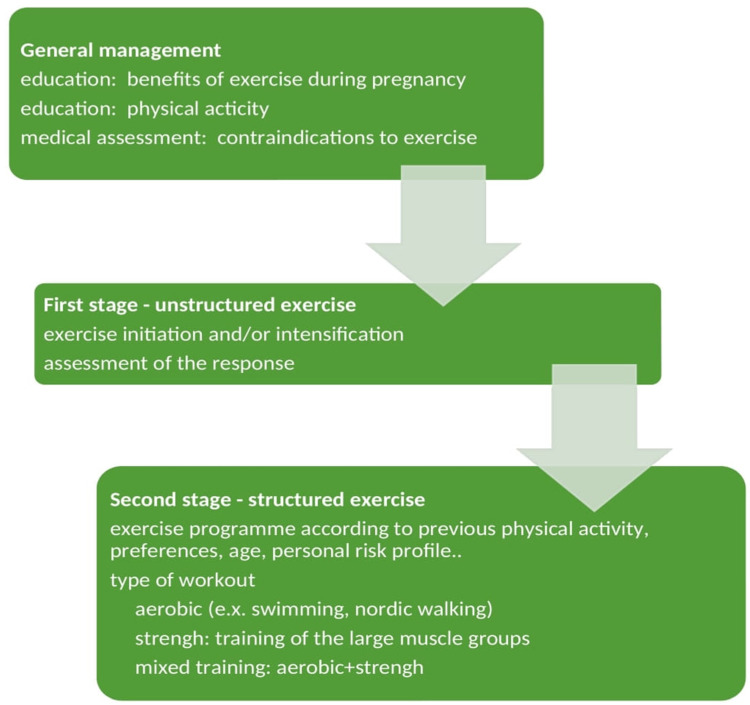
The principles of introducing and expanding physical exercise in pregnancy.

**Table 1 ijerph-20-01267-t001:** Pregnancy risks and consequences for the baby in overweight and obese women.

Mother	Fetus/Newborn/Child
Early pregnancy loss	Congenital malformations
GDM	Macrosomia—LGA newborn, increased adiposity
Hypertensive disorders	Shoulder dystocia—perinatal injuries
Preterm birth	Fetal asphyxia
Anesthesia complications	Stillbirth
Operative delivery—cesarean delivery	
Intrapartum hemorrhage	Obesity
Thromboembolic disease	Metabolic syndrome
Depression	Cardiovascular diseases
	Cognitive disorders

**Table 2 ijerph-20-01267-t002:** Recommended weight gain during pregnancy.

BMI Before Conception (kg/m^2^)	Weight Gain During Pregnancy (kg)
<18.5	13–18
18.5–24.9	11–16
25.0–29.9	7–11
≥30	5–9

**Table 3 ijerph-20-01267-t003:** Selected meta-analyses on the effect of exercise on PE prevalence.

Authors	Studied Groups	Number of Participants	Impact on PE, GH or GHDs	Additional Information
Syngelaki et al. [330]	Obese and overweight pregnant women	*n* = 1271	PE RR 1.01 95% CI 0.80–1.27, NS GH RR 0.87 95% CI 0.70–1.06, NS	
Magro-Malosso et al. [325]		PE *n* = 2230 GH *n* = 4641 GHDs *n* = 2517	PE RR 0.79 95% CI 0.45–1.38, NS GH RR 0.54 95% CI 0.40–0.74 GHDs RR 0.70 95% CI 0.53–0.83	Women assigned before 23rd week; Aerobic exercise 30–60 min., 2–7 times a week
Davenport et al. [331]		PE *n* = 3322 GH *n* = 5316	PE RR 0.59 95% CI 0.37- 0.9 GH RR 0.61 95% CI 0.43–0.85	Exercise interventions: the frequency of exercise ranged from 1 to 7 days/week; the duration of exercise ranged from 10 to 90 min. per session. No studies looked at the effect of exercising in different trimesters on the odds of developing GH or PE. Patients included in the study throughout pregnancy. To achieve at least a 25% reduction in the odds of developing PE and GH, pregnant women need to accumulate at least 600 MET-min/week of moderate-intensity exercise. Benefits would be attained when exercise is performed at a frequency of at least 3 days/week or at least 25 min. per session.
Brown et al. [332]	Patients with GDM	PE *n* = 48	RR 0.31 95% CI 0.01–7.09, NS	
Du et al. [333]	Obese and overweight pregnant women	PE *n* = 596 GH *n* = 671	PE RR 1.39 95% CI 0.66–2.93, NS GH RR 0.63 95% CI 0.38–1.05, NS	Patients included in the studies before 20th week. Exercise programs in the intervention groups were “stationary cycling” at least 3 times/week (30 min./session); “aerobic strength and muscle exercises”: 2–3 times/week (60 min./session); “walking for 11,000 steps or at least 30 min./daily; “mixed method” of stationary cycling, tread mill walk and muscle exercise”: 1–3 times/week (about 50 min./session); and “personalized exercise plan”.
Adesegun et al. [334]	Patients with pre-gestational diseases, including chronic hypertension, type 1 and 2 diabetes	PE *n* = 109	RR 0.80 95% CI 0.27–2.40, NS	Exercise during pregnancy.
Rogozińska et al. [335]		PE and PIH *n* = 8852	RR 0.95 95% CI 0.78–1.16, NS	
Zheng et al. [336]		*n* = 507	RR 1.05 95% CI 0.53–2.07, NS	Exercise between 10 and 20th week.
Aune et al. [323]		Pre-pregnancy *n* = 621	Pre-pregnancy RR 0.65 95% CI 0.47–0.89 RR 0.72 95% CI 0.53–0.99; per 1 h per day; RR 0.78 95% CI 0.63–0.96; per 20 MET-hours per week.	Dose–response analyses included.
		Early pregnancy *n* = 5702	Early pregnancy RR 0.79 95% CI 0.70–0.91 RR 0.83 95% CI 0.72–0.95; per 1 h per day RR 0.85 95% CI 0.68–1.07; per 20 MET-hours per week.	Defined as up to 24th week, before first prenatal visit or during 1st trimester.
Muhammad et al. [337]	Obese and overweight pregnant women	PE and/or PIH *n* = 650	RR 0.77 95% CI 0.46–1.30, NS	Supervised exercise.
Muktabhant et al. [310]	PE—mixed and high-risk population	PE Supervised exercise *n* = 1024 Unsupervised exercise *n* = 229 Maternal hypertension (not prespecified) Supervised exercise *n* = 1749 Unsupervised exercise *n* = 229	PE RR 0.95 95% CI 0.77–1.16, NS RR 1,6 95% CI 0.38–6.73, NS Maternal hypertension RR 0.55 95% CI 0.29–1.03 RR 0.43 95% CI 0.12–1.54	Supervised and unsupervised exercise.
Xing et al. [338]	Obese and overweight pregnant women	PE *n* = 1004	RR 1.00 95% CI 0.66–1.52, NS	
da Silva et al. [309]		PE *n* = 709	RR 0.93 95% CI 0.55–1.57, NS	Leisure-time physical activity throughout pregnancy.
Teede et al. [326]		PE and GH *n* = 5332	RR 0.66 95% CI 0.48–0.90	Structured exercise.

PE—preeclampsia, GH—gestational hypertension, PIH—pregnancy-induced hypertension, GHDs—gestational hypertensive disorders, GDM—gestational diabetes mellitus, NS—not significant, RR—relative risk, CI—confidence interval, MET—metabolic equivalent.

**Table 4 ijerph-20-01267-t004:** Safe and unsafe exercise during pregnancy.

Safe Physical Activity	Unsafe Physical Activity
Walking Swimming Stationary cycling Low-intensity aerobics Yoga, modified * Pilates, modified * Only for pregnant women with an uncomplicated course of pregnancy, after consultation with an obstetrician	Contact sports (e.g., basketball and volleyball) Exercise with a high risk of falling (e.g., skiing, riding, and cycling) Yoga, pilates, and stationary cycling in extreme conditions (high ambient temperature)

* In a position other than on the back.

**Table 5 ijerph-20-01267-t005:** Contraindications to exercise in pregnancy.

Absolute Contraindications	Relative Contraindications
Severe cardiovascular and respiratory diseases Severe anemia Increased risk of premature labor: incompetent cervix, cervical cerclage, multiple gestation Symptoms of preterm labor: preterm premature rupture of membranes, uterine contractions Vaginal bleeding Pre-eclampsia or gestational hypertension	Anemia Undiagnosed maternal cardiac arrhythmias Chronic bronchitis Poorly controlled diabetes mellitus type 1 Extreme morbid obesity or extreme underweight Fetal growth restriction during current pregnancy Poorly controlled chronic diseases: chronic hypertension, hyperthyroidism, seizure disorder Significant musculoskeletal disorders

## Data Availability

The data used to support the findings of this study are included in the article.

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
