# Peer review of "Preeclampsia and Obesity—The Preventive Role of Exercise"

_ijerph, 2023, doi:10.3390/ijerph20021267_

Round 1
Reviewer 1 Report
Preeclampsia and obesity. Role of exercise.
This is well-written paper with evidently a great effort and a lot of hours spent to write it. The authors wanted to make an encyclopedic paper on preeclampsia (with 342 references).
However « this article aims to present the mechanisms of the development of hypertension in pregnancy in obese women and the importance of physical activity in its prevention ». Page 3, line 107-8.
Then follows 6 pages (pages 3 to 8) on pathophysiology of preeclampsia (reaching 114 references) leading the reader to spend some 30-45 minutes before reaching the core of the paper (and the title’s goal, for what he decided to read the article) : 3. Obesity and preeclampsia. Page 8
These 6 pages and 114 references are worth to be a paper by itself, and I would recommend to the authors to submit it in a different article (why not in this same Journal). There, many comments could be done like the core- reflection on oxidative stress which may be challenged
Huppertz B. Oxygenation of the placenta and its role in pre-eclampsia. Pregnancy Hypertens. 2014 Jul;4(3):244-5. doi: 10.1016/j.preghy.2014.04.016. Epub 2014 Jul 9. PMID: 26104644.
Huppertz B, Weiss G, Moser G. Trophoblast invasion and oxygenation of the placenta: measurements versus presumptions. J Reprod Immunol. 2014 Mar;101-102:74-79. doi: 10.1016/j.jri.2013.04.003. Epub 2013 Jun 6. PMID: 23747129.
Or, page 4, lines 186 to 190 « the causes ….One of them is supposed to be an impaired maternal immune response against the allogeneic fetus….. In physiological pregnancy….an immunosuppressed state prevails » (references from 2006).
In immunology of preeclampsia, this no more the current consensus :
[Robillard PY, Dekker G, Scioscia M, Saito S. Progress in the understanding of the pathophysiology of immunologic maladaptation related to early-onset preeclampsia and metabolic syndrome related to late-onset preeclampsia. Am J Obstet Gynecol. 2022 Feb;226(2S):S867-S875. doi: 10.1016/j.ajog.2021.11.019. Epub 2022 Jan 5. PMID: 35177223.]
A physiological pregnancy is absolutely not an immunosuppressed state. It is a very ACTIVE immunological state of recognition of the alloggeneic fetus. It is the failure of this active state of recognition (to induce immunological tolerance) which leads to a kind of hemi-graft rejection, and therefore a bad trophoblastic invasion (early onset preeclampsia, IUGR).
I would do 1. Introduction , 2.. Obesity and preeclampsia.
1.INTRODUCTION. Very well presented, defining the problem properly. I would add in this introduction Line 48 « prevention of PE is acetylsalicylic acid AA [3,6-8], but this prevention is valid only for early onset preeclampsia < 34 weeks gestation [Rolnik]
Rolnik DL, Nicolaides KH, Poon LC. Prevention of preeclampsia with aspirin. Am J Obstet Gynecol. 2022 Feb;226(2S):S1108-S1119. doi: 10.1016/j.ajog.2020.08.045. Epub 2020 Aug 21. PMID: 32835720.
As a matter of fact, there is now an universal consensus to differentiate between early onset preeclampsia (EOP < 34 weeks gestation) and late onset preeclampsia (LOP, 34 weeks onward).
Tranquilli AL, Brown MA, Zeeman GG, Dekker G, Sibai BM. The definition of severe and early-onset preeclampsia. Statements from the International Society for the Study of Hypertension in Pregnancy (ISSHP).Pregnancy Hypertens. 2013 ;3(1):44-7. doi: 10.1016/j.
But, LOP represents by far the major component of preeclampsia : 90% of cases of PE and AA prevention is inefficient for this predominent form.
Major epidemiological evidences have shown recently in different populations : Reunion island, SCOPE study (Australia, New-Zealand , UK ..) [Robillard PY, Dekker G, Scioscia M, Bonsante F, Iacobelli S, Boukerrou M, Hulsey TC. Increased BMI has a linear association with late-onset preeclampsia: A population-based study. PLoS One. 2019 Oct 17;14(10):e0223888. doi: 10.1371/ journal.pone.0223888. PMID: 31622409; PMCID: PMC6797165.]
and in a National US data base [Bicocca MJ, Mendez-Figueroa H, Chauhan SP, Sibai BM. Maternal Obesity and the Risk of Early-Onset and Late-Onset Hypertensive Disorders of Pregnancy. Obstet Gynecol. 2020 Jul;136(1):118-127. doi: 10.1097/AOG.0000000000003901. PMID: 32541276.]
that incidence of LOP was linearly associated with increasing BMIs and much more poorly with EOP. Prevention of EOP is achievable with aspirin prevention [Rolnik], but the fact that the highly predominent LOP is specifically associated with pre-pregnancy overweight and different kinds of obesities (increasing linearly between obesities class I to III) allows to think that LOP is rather linked with metabolic syndrome (in contrary with EOP).
As a matter of fact, interventions on gestational weight gain planned since the beginning of a 9 month pregnancy has the the potential to halve the incidence of LOP in overweight/obese women
[Robillard PY, Dekker G, Boukerrou M, Boumahni B, Hulsey T, Scioscia M. Gestational weight gain and rate of late-onset preeclampsia: a retrospective analysis on 57 000 singleton pregnancies in Reunion Island. BMJ Open. 2020 Jul 28;10(7):e036549. doi: 10.1136/bmjopen-2019-036549. PMID: 32723741; PMCID: PMC7389512.]
In this view, active intervention on obesity during pregnancy makes sense to counterbalance its morbid effect (especially on the incidence of late onset preeclampsia. « this article aims to present the mechanisms of the development of hypertension in pregnancy in obese women and the importance of physical activity in its prevention ».
2. Obesity and preeclampsia.
The authors should add a paragraph 3.7 on d-chiro inositol phosphoglycan (after 3.6 Insulin resistance)
Scioscia M, Nigro M, Montagnani M. The putative metabolic role of d-chiro inositol phosphoglycan in human pregnancy and preeclampsia. J Reprod Immunol. 2014 Mar;101-102:140-147. doi: 10.1016/j.jri.2013.05.006. Epub 2013 Aug 2. PMID: 23962711.
Scioscia M, Noventa M, Cavallin F, Straface G, Pontrelli G, Fattizzi N, Libera M, Rademacher TW, Robillard PY. Exploring strengths and limits of urinary D-chiro inositol phosphoglycans (IPG-P) as a screening test for preeclampsia: A systematic review and meta-analysis. J Reprod Immunol. 2019 Sep;134-135:21-27. doi: 10.1016/j.jri.2019.07.005. Epub 2019 Jul 26. PMID: 31382126.
Scioscia M, Karumanchi SA, Goldman-Wohl D, Robillard PY. Endothelial dysfunction and metabolic syndrome in preeclampsia: an alternative viewpoint. J Reprod Immunol. 2015 Apr;108:42-7. doi: 10.1016/j.jri.2015.01.009. Epub 2015 Feb 21. PMID: 25766966.
DISCUSSION.
Around line 911 « that classes of obesity should be considered the 3 following references should be re-cited [Robillard et al 2019, Bicocca et al 2020]
Very good work, congratulations

Author Response
Thank you very much for your insightful review and valuable comments.
We do appreciate your great commitment to improving our work.
Please allow us to respond to your comments.
This is well-written paper with evidently a great effort and a lot of hours spent to write it. The authors wanted to make an encyclopedic paper on preeclampsia (with 342 references). However « this article aims to present the mechanisms of the development of hypertension in pregnancy in obese women and the importance of physical activity in its prevention ». Page 3, line 107-8.
Then follows 6 pages (pages 3 to 8) on pathophysiology of preeclampsia (reaching 114 references) leading the reader to spend some 30-45 minutes before reaching the core of the paper (and the title’s goal, for what he decided to read the article) : 3. Obesity and preeclampsia. Page 8
These 6 pages and 114 references are worth to be a paper by itself, and I would recommend to the authors to submit it in a different article (why not in this same Journal). There, many comments could be done like the core- reflection on oxidative stress which may be challenged
Huppertz B. Oxygenation of the placenta and its role in pre-eclampsia. Pregnancy Hypertens. 2014 Jul;4(3):244-5. doi: 10.1016/j.preghy.2014.04.016. Epub 2014 Jul 9. PMID: 26104644.
Huppertz B, Weiss G, Moser G. Trophoblast invasion and oxygenation of the placenta: measurements versus presumptions. J Reprod Immunol. 2014 Mar;101-102:74-79. doi: 10.1016/j.jri.2013.04.003. Epub 2013 Jun 6. PMID: 23747129.
We are highly grateful for your comments on this section of our manuscript. The decision to include the chapter on the pathophysiology of preeclampsia in this expanded form was based on several considerations. The scope of the IJERPH as well as this special issue is very broad, hence in addition to obstetricians, for whom it may be a summary of the current state of knowledge on the pathophysiology of preeclampsia, we are aware that the audience of this work may also include specialists in other medical fields. We believe that these readers will benefit from the presentation of mechanisms involved in developing preeclampsia. It allows them to understand why physical activity, at least theoretically, can prevent the development of PE.
We are extremely pleased that, in the reviewer's opinion, this part of the paper could form the basis of a separate publication. Such a publication would include an in-depth description of the mechanisms responsible for developing PE, including oxidative stress or immune abnormalities. We will deeply consider this suggestion.
Or, page 4, lines 186 to 190 « the causes ….One of them is supposed to be an impaired maternal immune response against the allogeneic fetus….. In physiological pregnancy….an immunosuppressed state prevails » (references from 2006).
In immunology of preeclampsia, this no more the current consensus :
[Robillard PY, Dekker G, Scioscia M, Saito S. Progress in the understanding of the pathophysiology of immunologic maladaptation related to early-onset preeclampsia and metabolic syndrome related to late-onset preeclampsia. Am J Obstet Gynecol. 2022 Feb;226(2S):S867-S875. doi: 10.1016/j.ajog.2021.11.019. Epub 2022 Jan 5. PMID: 35177223.]
A physiological pregnancy is absolutely not an immunosuppressed state. It is a very ACTIVE immunological state of recognition of the allogeneic fetus. It is the failure of this active state of recognition (to induce immunological tolerance) which leads to a kind of hemi-graft rejection, and therefore a bad trophoblastic invasion (early onset preeclampsia, IUGR).
This excerpt has been changed and the mentioned article has been quoted.
I would do 1. Introduction , 2.. Obesity and preeclampsia.
We do believe that our explanation for leaving the chapter on the pathophysiology of PE will find recognition and understanding from the Reviewer.
1.INTRODUCTION. Very well presented, defining the problem properly. I would add in this introduction Line 48 « prevention of PE is acetylsalicylic acid AA [3,6-8].
The introduction was supplemented with the suggested paragraph and references.
- Obesity and preeclampsia.
The authors should add a paragraph 3.7 on d-chiro inositol phosphoglycan (after 3.6 Insulin resistance).
The text was supplemented with a suggested paragraph and citations.
DISCUSSION.
Around line 911 « that classes of obesity should be considered the 3 following references should be re-cited [Robillard et al 2019, Bicocca et al 2020]
Suggested references have been re-cited.
We would like to sincerely thank you for your very insightful, valuable and constructive comments and for your effort and time spent evaluating our article. We do hope that applying the suggested changes will make it possible to publish our article.
Reviewer 2 Report
This is a clearly written and well-organized review. The authors have addressed elaboratively preeclampsia and obesity and how exercise can be used as a preventive measure to manage maternal and fetal health. Preeclampsia is one of the common conditions occurring in 5-8% of pregnancies all over the world. In the United States, it is one of the causes of 15% of premature births. The paper intends to show the significance of exercise in targeting obesity, a pro-inflammatory state to reduce both the early and late- onset of preeclampsia. The paper gives detailed information on how exercise can be used as a prophylactic measure during pregnancy to prevent pregnancy complications. Having that said, the article needs minor revision before it is published.
The comments are mentioned below:
P2, line 67: Definition of BMI, it should be body weight and height.
P6, line 255: VEGFR1 is also known as frns like tyrosine kinase-1 (Flt-1). It should be Flt-1 not sFlt-1 as sFlt-1 is a decoy receptor.
P7, line 313: It should be PPAR isoforms, not PAPPR.
P7, line 326: It should be PPARγ, not PPARy
P16, line 791: Change the sentence to “may justify their use in the prevention of PE”.
P17, line 831: Authors should correct the sentence.
P18, line 861: Figure 2: It should be VCAM-1
P21, line 999: Table 3: Authors should mention if the meta-analyses was done in primipara pregnancy, exercise details, and the subjects included in these studies.
P25, Table 4: Authors should correct the grammar mistakes in the table.
Author Response
Thank you very much for your insightful review and valuable comments.
We do appreciate your great commitment to improving our work.
Please allow us to respond to your comments:
P2, line 67: Definition of BMI, it should be body weight and height.
It has been corrected.
P6, line 255: VEGFR1 is also known as frns like tyrosine kinase-1 (Flt-1). It should be Flt-1 not sFlt-1 as sFlt-1 is a decoy receptor.
It has been corrected.
P7, line 313: It should be PPAR isoforms, not PAPPR.
It has been corrected.
P7, line 326: It should be PPARγ, not PPARy
It has been corrected.
P16, line 791: Change the sentence to “may justify their use in the prevention of PE”.
It has been corrected
P17, line 831: Authors should correct the sentence.
It has been corrected
P18, line 861: Figure 2: It should be VCAM-1
It has been corrected
P21, line 999: Table 3: Authors should mention if the meta-analyses was done in primipara pregnancy, exercise details, and the subjects included in these studies.
Unfortunately, not all meta-analyses provide detailed information on the included patients' characteristics and exact data on the type and intensity of exercise.
It makes it significantly challenging to compare the results of these meta-analyses. Hence, the table includes all data provided by the cited meta-analyses.
P25, Table 4: Authors should correct the grammar mistakes in the table.
The corrections, after language consultation, have been made.
We would like to sincerely thank you for your very insightful, valuable and constructive comments and for your effort and time spent evaluating our article. We do hope that applying the suggested changes will make it possible to publish our article.